# Different contributions of preparatory activity in the basal ganglia and cerebellum for self-timing

Jun Kunimatsu[1,2]*, Tomoki W Suzuki[1], Shogo Ohmae[1,3], Masaki Tanaka[1]*

[1]Department of Physiology, Hokkaido University School of Medicine, Sapporo, Japan; [2]Laboratory of Sensorimotor Research, National Eye Institute, National Institutes of Health, Bethesda, United States; [3]Department of Neuroscience, Baylor College of Medicine, Houston, United States

**Abstract** The ability to flexibly adjust movement timing is important for everyday life. Although the basal ganglia and cerebellum have been implicated in monitoring of supra- and sub-second intervals, respectively, the underlying neuronal mechanism remains unclear. Here, we show that in monkeys trained to generate a self-initiated saccade at instructed timing following a visual cue, neurons in the caudate nucleus kept track of passage of time throughout the delay period, while those in the cerebellar dentate nucleus were recruited only during the last part of the delay period. Conversely, neuronal correlates of trial-by-trial variation of self-timing emerged earlier in the cerebellum than the striatum. Local inactivation of respective recording sites confirmed the difference in their relative contributions to supra- and sub-second intervals. These results suggest that the basal ganglia may measure elapsed time relative to the intended interval, while the cerebellum might be responsible for the fine adjustment of self-timing.

DOI: https://doi.org/10.7554/eLife.35676.001

**\*For correspondence:**
kunimatsu.jun@gmail.com (JK);
masaki@med.hokudai.ac.jp (MT)

**Competing interests:** The authors declare that no competing interests exist.

## Introduction

Action timing is crucial for organisms to interact with dynamic environments. To make timely movements, elapsed time often needs to be monitored and the timing of upcoming events needs to be accurately predicted. Previous studies have shown that, in addition to the cerebral cortex, both the basal ganglia and the cerebellum are implicated in this function (*Buhusi and Meck, 2005*). In humans, the underlying neural mechanism of self-timing has been examined by recording scalp potentials that develop gradually over the medial frontal cortex in anticipation of task-relevant events (contingent negative variation or CNV: *Macar and Vidal, 2003*; *van Rijn et al., 2011*). However, CNV sometimes provides only a poor indication of event timing (*Kononowicz and van Rijn, 2011*), possibly because it contains multiple components. In fact, the magnitudes of CNV are known to be correlated with neural activity in the thalamus, the basal ganglia, and the cerebellum (*Nagai et al., 2004*), indicating that the ramping-up of neuronal activity in the cerebral cortex is likely to be regulated by different signals arising from multiple subcortical structures.

More direct evidence for the role of subcortical signals in motor preparation has come from animal experiments. In monkeys, neurons in the motor thalamus exhibit a gradual rise in firing rate that predicts the timing of self-initiated movements (*Costello et al., 2016*; *Tanaka, 2007*), and their inactivation causes delayed self-timing (*Tanaka, 2006*; *van Donkelaar et al., 2000*). Recent studies in rodents have demonstrated that direct inputs from the thalamus are necessary for generating preparatory activity in the premotor cortex (*Guo et al., 2017*), while signals in the cortical network are also essential for generation (*Murakami et al., 2014*) and maintenance (*Li et al., 2016*) of ramping activity. Because the thalamus transmits signals from both the basal ganglia and the cerebellum,

elucidation of the neuronal activity in these subcortical structures will address specific roles of respective cortico-subcortical loops in self-timing.

A prevailing hypothesis regarding subcortical contribution to timing is that the basal ganglia process supra-second interval timing while the cerebellum is involved in sub-second timing (*Ivry and Spencer, 2004*). Our recent analysis in monkeys showed that neurons in the cerebellar dentate nucleus exhibited preparatory activity only about a half second before self-initiated saccades, irrespective of the length of the mandatory delay interval that ranged from 400 to 2400 ms (*Ohmae et al., 2017*). These results indicate that neurons outside of the cerebellum must keep track of elapsed time to make temporally accurate movements in trials with supra-second delay intervals. Because neurons in the striatum are known to exhibit ramping activity during motor preparation (*Schultz and Romo, 1992*), signals in the cortico-basal ganglia loop might play this role.

To understand the roles of the basal ganglia and the cerebellum in self-timing, we compared activity of single neurons in the anterior part of the striatum (caudate nucleus) with those in the posterior part of the cerebellar dentate nucleus in monkeys performing the self-timed saccade task (*Ashmore and Sommer, 2013*; *Kunimatsu and Tanaka, 2012*, *2016*; *Tanaka, 2006*, *2007*). Because the previous studies have demonstrated that these subcortical regions have common anatomical connections with the medial and lateral frontal cortices involved in temporal processing (*Dum and Strick, 2003*; *Strick et al., 2009*; *McFarland and Haber, 2001*), we reasoned that these areas might have functional interactions. Although the previous studies of eye blink conditioning suggest that plasticity in the cerebellar cortex rather than the deep cerebellar nuclei plays a role in the learning of motor timing (*Garcia and Mauk, 1998*; *Perrett et al., 1993*; *Raymond et al., 1996*), neurons in the nuclei encode timing signals originated in the cerebellar cortex (*Ten Brinke et al., 2017*). Therefore, we explored signals in the dentate nucleus in this study because any significant computation in the lateral cerebellum must modify neuronal activity in the nucleus that may eventually regulate movement timing. We found that neurons in the striatum displayed a ramping-up of firing rate throughout the delay period, while the rate of rise of neuronal activity depended on the length of the mandatory delay interval. In contrast, neurons in the cerebellar dentate nucleus exhibited preparatory activity only during the last part of the delay period, while neuronal correlates of trial-by-trial variation of saccade latency started earlier in the cerebellum than the striatum. These results suggest that the striatum may play a role in monitoring the passage of time relative to the mandatory interval, while the cerebellum might play a role in the fine adjustment of self-timing in the range of hundreds of milliseconds.

## Results

### Time courses of preparatory activity for self-timing in the striatum and the cerebellum

We examined neuronal activity while two Japanese monkeys performed the self-timed saccade task (*Figure 1A*). In this task, the animals made a self-initiated memory-guided saccade to the location of the previously presented brief visual cue (100 ms). They received a liquid reward for saccades generated within a predetermined time interval that was indicated by color of the fixation point (FP) in each trial. Distributions of saccade latency during recording sessions shown in *Figure 1B* (*Figure 1—source data 1*) indicate that both monkeys flexibly adjusted movement timing depending on the given instructions.

We recorded from 162 task-related neurons in the anterior part of the caudate nucleus (63 and 99 neurons for monkeys B and G, respectively) and 127 neurons in the posterior part of the cerebellar dentate nucleus (59 and 68, respectively, *Figure 2*). Of these, 100 striatal neurons (37 and 63 neurons for monkeys B and G, respectively) and 76 cerebellar neurons (29 and 47 neurons) showed elevated activity before self-timed saccades (see Materials and methods). Most of them exhibited increased activity for both saccade directions (55 and 66% for neuron in the caudate nucleus and the cerebellar dentate nucleus, respectively), while many showed a significant directional modulation (48/100 and 32/76, respectively; Wilcoxon rank sum test, p<0.05).

*Figure 3A* shows a representative example of caudate neuron exhibiting a gradual ramp-up of firing rate before self-timed saccades for all three mandatory delay intervals. This neuron elevated activity immediately after cue appearance but the rate of rise of firing rate differed for different

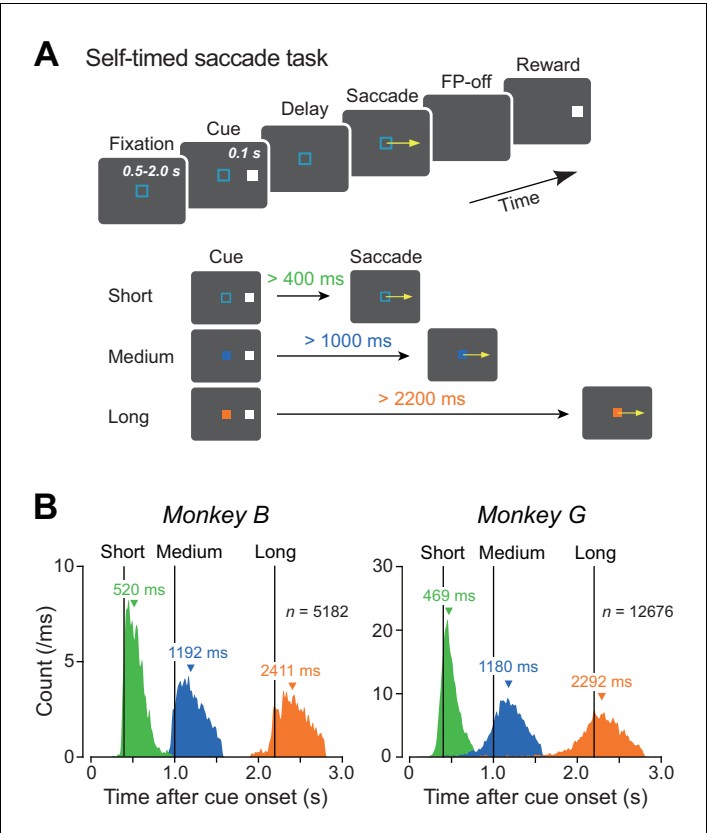

**Figure 1.** Behavioral task and performance. (**A**) Sequence of events in the self-timed saccade task (upper panel). During central fixation, a cue flashed briefly (100 ms) in the peripheral visual field. Monkeys were required to remember the cue location and maintain fixation until expiration of the predetermined mandatory delay interval that was indicated by color of the fixation point (lower panel). Animals received a reward if they correctly made a self-timed memory-guided saccade to the cue location after the mandatory delay period. (**B**) Distributions of saccade latency during recording sessions in two monkeys. Differently colored histograms represent the data for different mandatory intervals. Vertical lines indicate the end of the mandatory intervals (400, 1000, and 2200 ms). Inverted triangles denote medians.

DOI: https://doi.org/10.7554/eLife.35676.002

The following source data is available for figure 1:

**Source data 1.** Data for *Figure 1B*.
DOI: https://doi.org/10.7554/eLife.35676.003

interval timing. In contrast, the representative cerebellar nuclear neuron shown in *Figure 3B* also exhibited a ramp-up activity, but the timing of ramp onset differed depending on the mandatory intervals, while the rate of rise of activity was roughly constant. The time courses of the population activity for each structure also gave a similar impression. Again, the ramp-up activity started just after cue onset in the striatum (*Figure 3C*, *Figure 3—source data 1*), while the firing modulation started late for trials with longer delay in the cerebellum (*Figure 3D*). The rate of rise of preparatory activity differed for different interval timing in the caudate nucleus (*Figure 3C*), while the activity was similar across all timing in the cerebellar dentate nucleus (*Figure 3D*). These results held consistent even when the same sets of data were aligned with saccade initiation (*Figure 3—figure supplement 1*). In general, the time courses of individual neuronal activity during the delay interval were more variable in the striatum than the cerebellum (*Figure 3—figure supplement 2*). Neurons in both structures did not show clear ramping activity during the standard memory-guided saccade task, in which animals generated a saccade in response to the FP offset (*Figure 3—figure supplement 3*). In addition to the delay period activity, the activity just before the cue onset in the self-timed task also tended to be different (striatum, 9.8 ± 5.6, 8.4 ± 9.2, 7.8 ± 8.6 spikes/s for short, medium and long

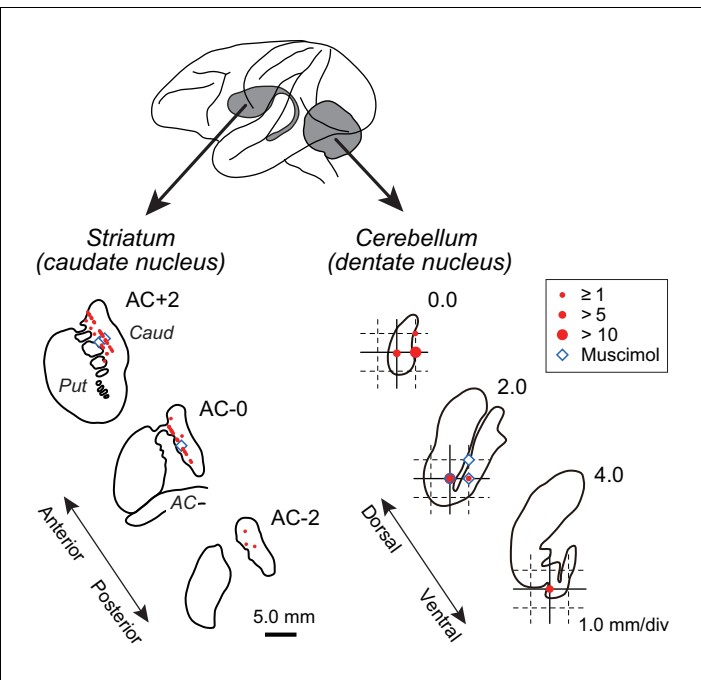

**Figure 2.** Recording and inactivation sites in monkey G. Drawings indicate coronal sections of the striatum (left panel) and horizontal sections of the cerebellar dentate nucleus (right panel). Red and blue symbols indicate the sites of recording and muscimol injection, respectively. Data for the intermediate sections are projected anteriorly (striatum) or dorsally (dentate nucleus). Note that the scales for the left and right panels differ by three times. AC, anterior commissure; Caud, caudate nucleus; Put, putamen.

DOI: https://doi.org/10.7554/eLife.35676.004

intervals, respectively; cerebellum, 50.6 ± 22.3, 52.0 ± 27.0, 46.9 ± 21.9 spikes/s), although these differences were not statistically significant (one-way ANOVAs, p=0.33 and 0.40).

To quantify the time courses of preparatory activity, the population of normalized activity in each condition was fit with a regression line incorporating a delay in ramp onset (*Figure 4A*, Materials and Methods). The left panel of *Figure 4B* summarizes the onset times of preparatory activity derived from the fitted lines and the distributions from bootstrap resampling for different conditions (*Figure 4—source data 1*). For all iterations (*n* = 1000), coefficients of determination ($r^2$) of fitting were greater than 0.86, and averaged 0.96 ± 0.01 (SD) for both structures. Except for the shortest delay interval, the times of ramp onset differed significantly between neurons in the striatum and those in the cerebellum (p<0.05, bootstrap CIs). The right panel of *Figure 4B* plots the slopes of ramp-up activity for different conditions. Although the slopes differed among the mandatory delay intervals for neurons in both the striatum and cerebellum (one-way ANOVA, p<0.05), the difference was much greater in the striatum. Consistent with the results of ramp onset, the slopes for the medium and long delay conditions again showed a significant difference between neurons in the striatum and those in the cerebellum (p<0.05, bootstrap CIs). We also obtained similar results when the same sets of data were aligned with saccade initiation (*Figure 4C*, *Figure 4—source data 1*). Thus, while striatal neurons were active throughout the delay period, dentate nuclear neurons were recruited only during the last part of the delay period in the self-timed task.

## Neuronal correlates of trial-by-trial variation in self-timing

We have shown that neurons in the striatum flexibly alter the rate of rise of ramping activity depending on the instruction. However, in addition to the imposed task rule, the timing of self-initiated movements can also vary depending on stochastic fluctuation in neuronal activity. In fact, many previous studies sought neuronal correlates of trial-by-trial variation in self-timing under fixed conditions (*Jazayeri and Shadlen, 2015*; *Maimon and Assad, 2006*; *Murakami et al., 2017*; *Ohmae et al., 2017*; *Soares et al., 2016*; *Tanaka, 2007*).

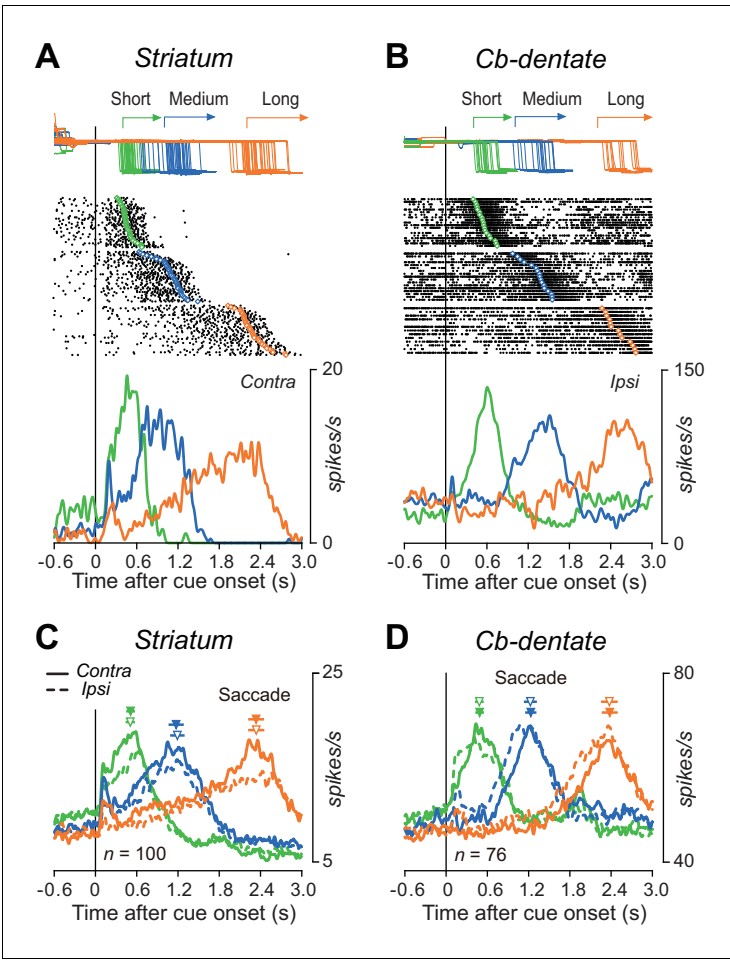

**Figure 3.** Comparison of single neuronal activity in the striatum and the cerebellar nucleus during the self-timed saccade task. (**A**) A representative neuron in the caudate nucleus showing a ramp-up of activity during the delay period. Trials are sorted by saccade latency, and the rasters and corresponding spike density are shown for saccades in contralateral direction. Green, blue, and orange traces indicate data for short, medium, and long mandatory intervals, respectively. (**B**) A representative neuron in the cerebellar dentate nucleus. Data are shown for saccades in ipsilateral direction. (**C**) Time courses of the population activity for neurons in the caudate nucleus. Traces are the means of spike densities for individual neurons and are aligned on the cue onset (vertical line). Continuous and dashed traces indicate data for saccades in contralateral and ipsilateral directions, respectively. The filled and open triangles with horizontal bars indicate the mean ± SD of average saccade latency for contralateral and ipsilateral directions in each session, respectively. (**D**) Time courses of the population activity for neurons in the dentate nucleus. The population data in both structures aligned with saccade initiation are shown in *Figure 3—figure supplement 1*.

DOI: https://doi.org/10.7554/eLife.35676.005

The following source data and figure supplements are available for figure 3:

**Source data 1.** Data for *Figure 3*.
DOI: https://doi.org/10.7554/eLife.35676.009
**Source data 2.** Data for *Figure 3—figure supplement 1*.
DOI: https://doi.org/10.7554/eLife.35676.010
**Source data 3.** Data for *Figure 3—figure supplement 2*.
DOI: https://doi.org/10.7554/eLife.35676.011
**Source data 4.** Data for *Figure 3—figure supplement 3*.
DOI: https://doi.org/10.7554/eLife.35676.012
**Figure supplement 1.** Time courses of the population activity aligned with saccade initiation.
DOI: https://doi.org/10.7554/eLife.35676.006
**Figure supplement 2.** Matrix of inter-neuronal correlation.
*Figure 3 continued on next page*

*Figure 3 continued*

DOI: https://doi.org/10.7554/eLife.35676.007

**Figure supplement 3.** Neuronal activity during the standard memory-guided saccade task and the visually-guided saccade task.

DOI: https://doi.org/10.7554/eLife.35676.008

To assess neuronal correlates of trial-by-trial variation, data for each neuron were divided into three groups according to saccade latency, and then the population of normalized activity was computed for each group. In the right panels in *Figure 5A and B*, the traces of spike density aligned on saccades are shifted in time so that the data alignments are placed at the mean saccade latencies relative to the cue onset (colored vertical lines, *Figure 5—source data 1*). For neurons in the striatum, preparatory activity started immediately after cue onset, but trial-by-trial variation was evident only during the very last part of the delay interval (*Figure 5A*). In contrast, for neurons in the cerebellar dentate nucleus, trial-by-trial variation started at the beginning of the preparatory activity except for trials with the shortest delay interval (*Figure 5B*).

We measured the timing of neuronal variation by comparing traces of individual neuronal activity among three groups for every millisecond (repeated measures ANOVA, $p < 0.01$, *Figure 5A and B*, inverted triangles). This procedure was optimal to detect the earliest time of diverging point of normalized spike density profiles for the population of neurons, but did not take account of the trial-by-trial fluctuation of baseline firing rate. Since the coefficient of variation of baseline activity was greater in the striatum than the cerebellum ($1.48 \pm 1.34$ vs. $0.52 \pm 0.29$, unpaired t-test, $p < 1.0E-8$), the estimate of timing for the striatal neurons could be much later if the variation of baseline firing was considered. *Figure 5C* summarizes the onset times of neuronal variation and the distributions from bootstrap resampling, computed for the data aligned with either the cue onset or saccade initiation (*Figure 5—source data 1*). The trial-by-trial variation of neuronal activity started earlier in the cerebellum than the striatum except for trials with the shortest mandatory delay interval ($p < 0.05$, bootstrap CIs). These results suggest that the cerebellum might be primarily responsible for the fine adjustment of self-timing, while the striatum may continuously monitor the time relative to the intended interval during the delay period.

## Duration preference in individual neurons

Several lines of evidence suggest that some neurons in the basal ganglia and the cerebral cortex might be tuned to specific interval timing (*Bartolo et al., 2014*; *Hayashi et al., 2015*; *Merchant et al., 2013b*; *Mita et al., 2009*; *Murakami et al., 2014*). Indeed, we found that a subset of caudate neurons showed a preference for specific mandatory delay interval, although the population activity at the time of saccade initiation was roughly the same across intervals (*Figure 3—figure supplement 1*). *Figure 6A* illustrates three such examples exhibiting ramp-up activity that was greatest for the short, medium, and long mandatory delay intervals.

To quantify the duration selectivity of preparatory activity in individual neurons, the firing rate 200 ms before self-timed saccades (black bars in *Figure 6A*) was compared across the delay intervals. Triangular plots in *Figure 6B* summarize the relative magnitude of the firing rate in individual neurons in the striatum and the cerebellum (*Figure 6—source data 1*). Among 31 (31%) caudate neurons exhibiting a significant duration selectivity (one-way ANOVA, $p < 0.01$), 18, 8, and five neurons showed a preference for short, medium, and long delay intervals, respectively. In contrast, a smaller population of neurons in the cerebellar dentate nucleus (11/77, 14%) showed significant duration selectivity, with a preference for short, medium, and long intervals in 1, 8, and two neurons, respectively. We obtained similar results for the data of ipsiversive (striatum) and contraversive (cerebellum) saccades (*Figure 6—figure supplement 1*). In both structures, we found no topographic organization of neurons with or without duration selectivity.

We also compared degree of firing modulation depending on interval timing. *Figure 6C* plots the cumulative distributions of the distance of each data point from the center of the triangular plots in *Figure 6B* (*Figure 6—source data 1*). For both structures, data for saccades in the opposite directions almost perfectly overlap (filled and open symbols). The graph clearly shows that striatal neurons tend to have greater duration selectivity than neurons in the cerebellar dentate nucleus (Kolmogorov-Smirnov test; contra, $p = 7.7E-11$; ipsi, $p = 5.3E-12$), whereas no obvious preference for a

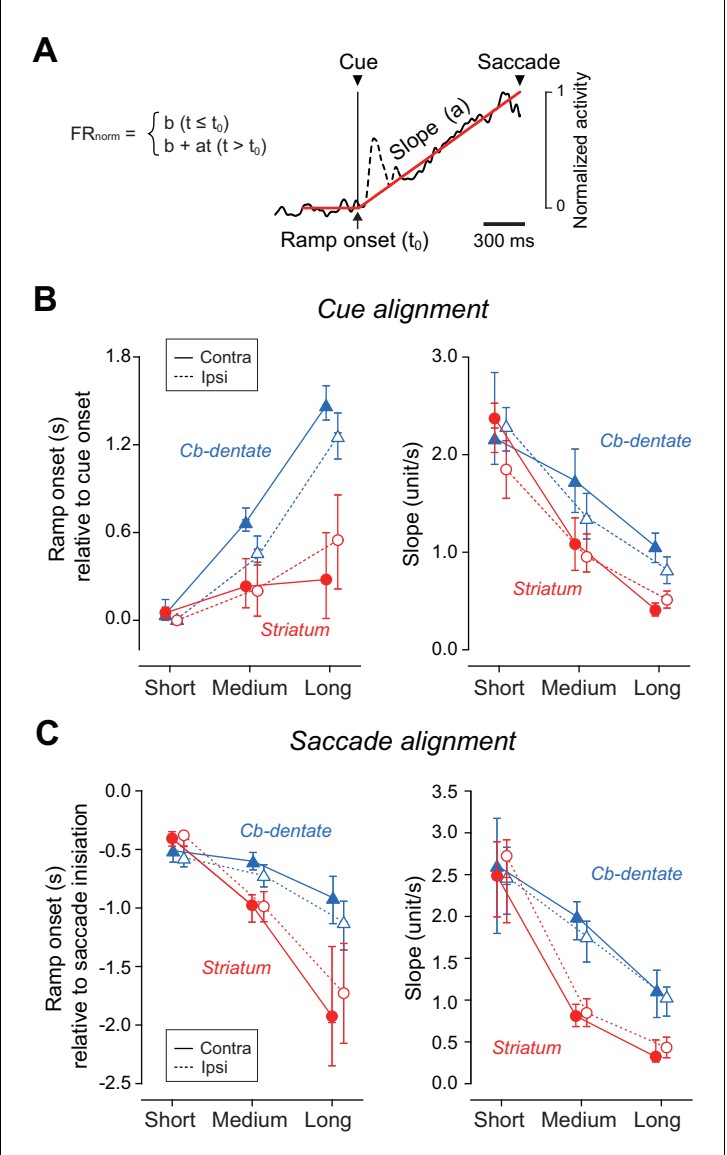

**Figure 4.** Quantitative analysis for the time course of ramping activity during the delay interval. (A) An example illustrating how we measured the onset and slope of ramp-up activity. For each condition, the trace of population of normalized activity (starting from 300 ms before cue onset and ending at the mean saccade latency) was fitted with two lines defined by three parameters (least squares, red lines). Data during 200 ms following 50 ms after cue onset were excluded to remove visual transients (dashed line). (B) Summary of ramp onset (left panel) and ramp-up slopes (right panel) for neurons in the striatum ($n$ = 100, red circles) and the cerebellar nucleus ($n$ = 76, blue triangles) based on data aligned with cue onset. Each data point indicates the values computed from the population data. Error bars with tick marks denote 2.5, 50 and 97.5 percentile of the results of the bootstrap analysis. Data points connected with solid and dashed lines indicate the data for ipsiversive and contraversive saccades, respectively. (C) Summary of ramp onsets (left panel) and slopes (right panel) computed for the data aligned with saccade initiation. Note that ramp onsets and slopes for the medium and long delay conditions differed significantly between the striatum and cerebellum.

DOI: https://doi.org/10.7554/eLife.35676.013

The following source data is available for figure 4:

**Source data 1.** Data for *Figure 4*.
DOI: https://doi.org/10.7554/eLife.35676.014

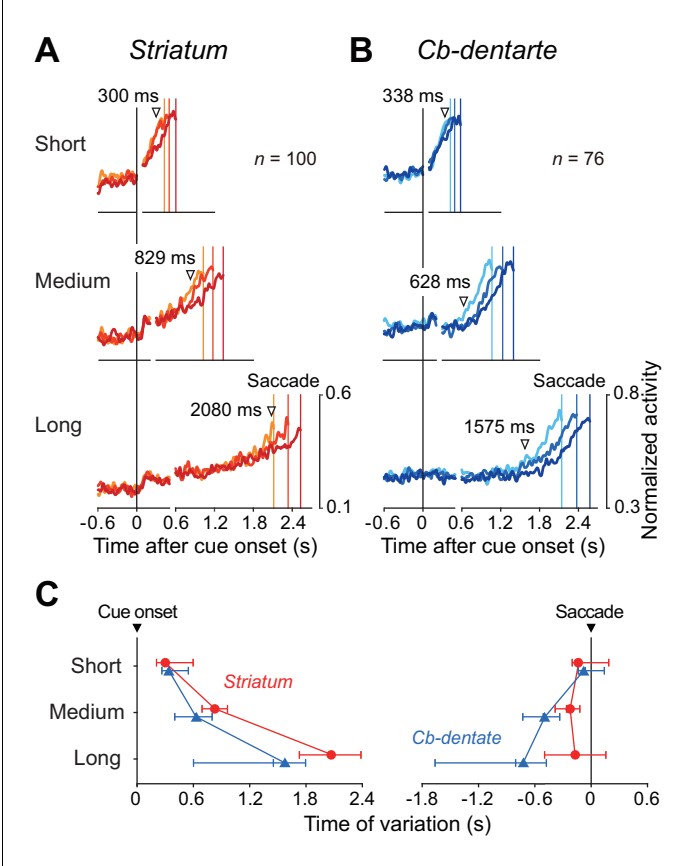

**Figure 5.** Timing of trial-by-trial variation of ramp-up activity. (**A, B**) For each condition, trials were divided into three groups according to saccade latency. Then, the data were normalized for each neuron, aligned with saccade initiation, and were shifted in time so that the times of saccades (colored vertical lines) were placed at the mean saccade latencies relative to the cue onset (right panels). On the left panels, data of the population activity were aligned with the cue onset (vertical black line). Inverted triangles indicate the time when the traces of normalized neuronal firing rate started to diverge as detected by repeated measures ANOVAs (p<0.01 for consecutive 40 ms). The baseline fluctuation (SD) of normalized neuronal activity was comparable between the recording sites (unpaired t-test, p=0.84). (**C**) Onset of trial-by-trial variation relative to the cue (left panel) or saccade initiation (right). Each point indicates the data derived from the analysis shown in (**A**) and (**B**). Error bars with three tick marks denote 2.5, 50 and 97.5 percentile of the results of the bootstrap analysis. Note that the trial-by-trial variation started earlier in the cerebellum than the striatum for medium and long intervals.
DOI: https://doi.org/10.7554/eLife.35676.015

The following source data is available for figure 5:

**Source data 1.** Data for *Figure 5*.
DOI: https://doi.org/10.7554/eLife.35676.016

specific duration was found in the population of neurons (*Figure 3—figure supplement 1* and *Figure 6B*).

## Inactivation effects on self-timing

To explore the causal role of neuronal activity, the recording sites in both monkeys were reversibly inactivated by injecting a small amount of muscimol. *Figure 7A* illustrates the cumulative latency distributions of self-timed saccades before and during inactivation of the caudate nucleus in a single experimental session. During inactivation, contraversive self-timed saccades were delayed in trials with short (400 ms) and long (2200 ms) mandatory delay intervals (Wilcoxon rank-sum test; short, p=7.4E-3; long, p=4.8E-3), while ipsiversive saccades were facilitated for all delay intervals (short, p=4.5E-3; medium, p=1.2E-7; long, p=2.0E-4). Among six experiments (three experiments for each

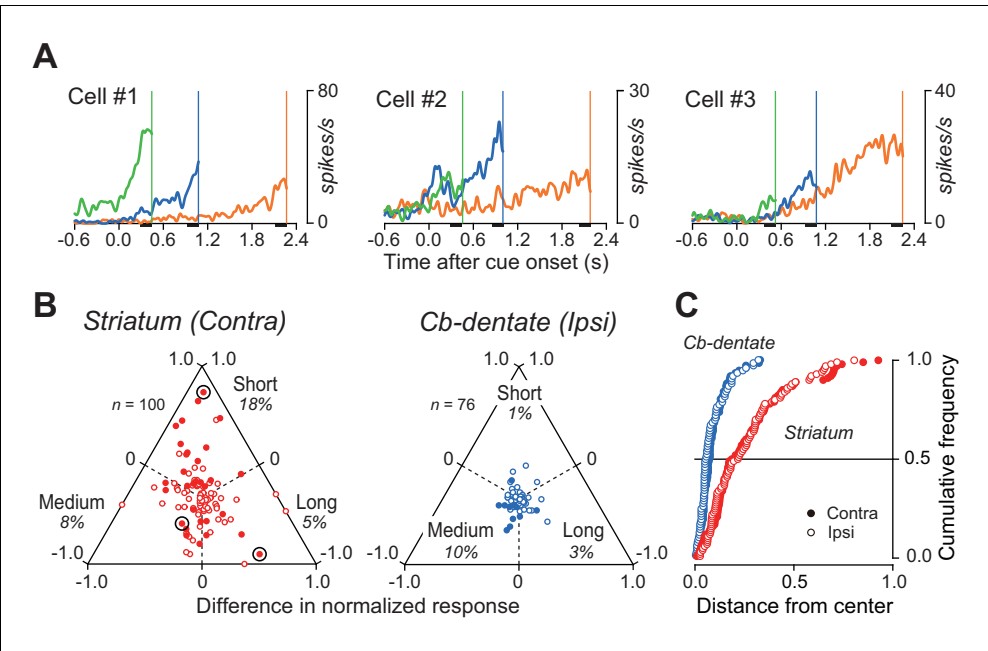

**Figure 6.** Duration preference of ramping activity. (**A**) Three striatal neurons exhibiting a preference for specific mandatory delay interval. For each neuron, data were aligned with self-timed saccades and were shifted in time to place the end of traces at the time of mean saccade latencies (vertical lines). (**B**) Comparison of the magnitude of firing modulation between the striatum and the cerebellar dentate nucleus. Filled symbols indicate the data showing a significant difference (ANOVA, p<0.01). Bull's eyes indicate the data for neurons shown in (**A**). Data for contraversive saccades only are shown for the striatum, while those for ipsiversive saccades only are shown for the cerebellum. Data for the opposite saccade directions are shown in *Figure 6—figure supplement 1*. Note that the data for the cerebellum clustered around the center, while those for the striatum varied. (**C**) Cumulative density functions for the distance from the center of the triangles for the data points in (**B**). Open and filled symbols indicate the data for saccades in ipsilateral and contralateral directions, respectively.

DOI: https://doi.org/10.7554/eLife.35676.017

The following source data and figure supplement are available for figure 6:

**Source data 1.** Data for *Figure 6*.
DOI: https://doi.org/10.7554/eLife.35676.019
**Source data 2.** Data for *Figure 6—figure supplement 1*.
DOI: https://doi.org/10.7554/eLife.35676.020
**Figure supplement 1.** Duration preference of ramping activity.
DOI: https://doi.org/10.7554/eLife.35676.018

monkey), contraversive saccades were significantly delayed in three experiments, while ipsiversive saccades were significantly facilitated in four experiments (Wilcoxon rank-sum test, p<0.05 with Bonferroni correction). In the population as a whole, the delay of contraversive self-timed saccades was found only in trials with a long delay interval (*Figure 7C*, paired t-test, p=0.038, *Figure 7—source data 1*), while the facilitation of ipsiversive saccades was consistently observed for all delay intervals (short, p=0.044; medium, p=0.048; long, p=0.021). We also found a significant change in contraversive saccade latency in the standard memory-guided saccade trials (paired t-test, p=0.012), but neither ipsiversive memory-guided saccades nor visually-guided saccades in both directions were altered during inactivation. During inactivation of the striatum, variation of self-timed saccade latencies significantly increased in two, one and one experiments in trials with short, medium and long delay intervals, respectively (F-test, p<0.05 with Bonferroni correction), while the variation decreased in one experiment for all delay intervals. In the population, the variation of saccade latency remained unchanged in all conditions (*Figure 7E*, paired t-test, p>0.09, *Figure 7—source data 1*). These results indicate that signals in the caudate nucleus regulate self-timing, especially in trials with a long delay interval.

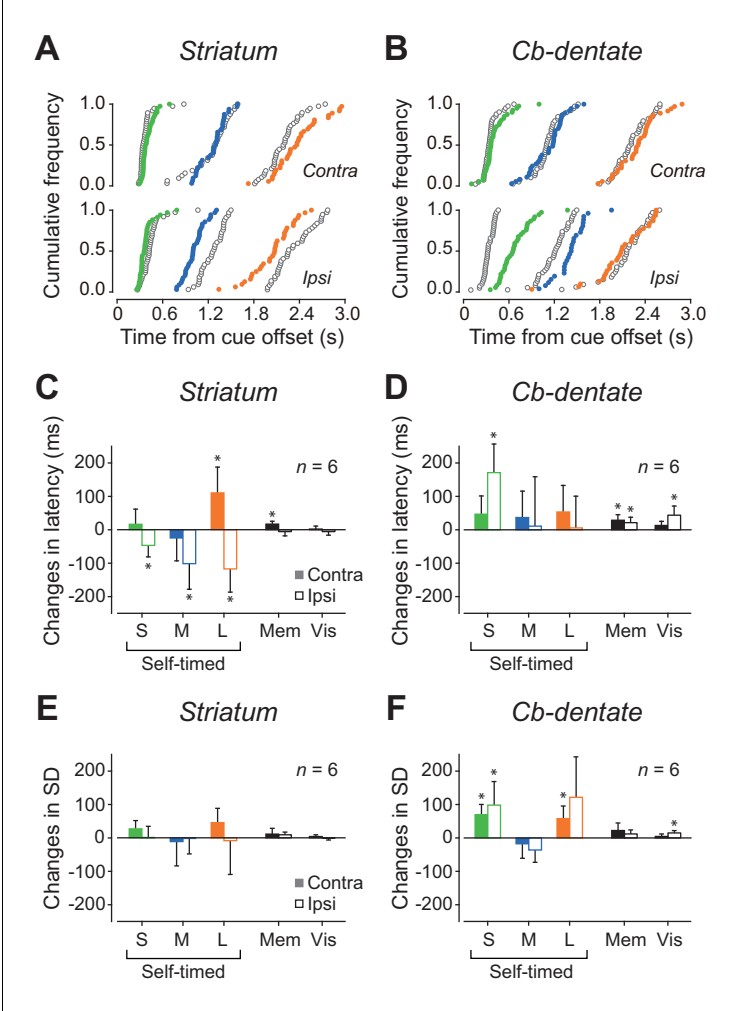

**Figure 7.** Effects of inactivation. (**A**) Data from a representative experiment in the caudate nucleus. Cumulative distributions of saccade latencies are compared between trials before (black open circles) and during (colored circles) inactivation with muscimol. Different colors indicate different mandatory delay intervals. (**B**) A representative experiment in the cerebellar dentate nucleus. Note that inactivation effects were greatest for ipsiversive trials with short mandatory intervals. (**C**) Summary of inactivation effects on saccade latency for the caudate nucleus. Bars and whiskers indicate the means and 95% confidence intervals of the changes in median latencies for different conditions. Filled and open bars indicate the data for contraversive and ipsiversive saccades, respectively. Asterisk denotes a significant inactivation effect (paired t-test, p<0.05). (**D**) Inactivation effects on saccade latency in the cerebellar nucleus. (**E, F**) Summary of inactivation effects on variation of saccade latency for the caudate nucleus and the cerebellar nucleus, respectively. SD, standard deviation.
DOI: https://doi.org/10.7554/eLife.35676.021

The following source data is available for figure 7:

**Source data 1.** Data for *Figure 7*.
DOI: https://doi.org/10.7554/eLife.35676.022

---

*Figure 7B* plots data from a single inactivation experiment performed in the cerebellum. Inactivation of the dentate nucleus delayed ipsiversive self-timed saccades in trials with short (400 ms) and medium (1000 ms) mandatory delay intervals (Wilcoxon rank-sum test; short, p=1.3E-9; medium, p=4.1E-3), while it failed to alter the timing of contraversive saccades. Among six experiments (three for each monkey), ipsiversive self-timed saccades were delayed in four and three experiments in trials with short and medium delay intervals, respectively. For contraversive saccades, changes in self-timing were found in two experiments in trials with short delay intervals (Wilcoxon rank-sum test,

p<0.05 with Bonferroni correction). In the population, a significant inactivation effect was found only for ipsiversive self-timed saccades with a short delay interval (*Figure 7D*, paired t-test, p=9.4E-3, *Figure 7—source data 1*). In addition, variation of ipsiversive self-timed saccade latencies significantly increased in four and two experiments in trials with short and long delay intervals, respectively. For contraversive saccades, the variation increased in four experiments in trials with short delay intervals (F-test, p<0.05 with Bonferroni correction). In the population as a whole, variation of self-timing increased for ipsiversive saccades with short delay interval (*Figure 7F*, paired t-test, p=0.041, *Figure 7—source data 1*) and contraversive saccades with short and long delay intervals (short, p=6.6E-3; long, p=0.030). In contrast, variation of saccade latency remained unchanged in the standard memory-guided saccade task. When we compared the coefficient of variation of self-timing, the values were statistically greater during inactivation than the control for ipsiversive saccades in trials with a short delay interval (paired t-test, p=0.042), indicating that the signals in the dentate nucleus may regulate the precision of self-timing in trials with a short delay interval. Because injections of saline into three effective sites in either structure failed to alter saccade latency in all conditions (Wilcoxon rank-sum test, p>0.05), the present results were not attributed to a non-specific volume effect. Taken together, the results of inactivation experiments were consistent with the notion that signals in the cerebellum mostly regulate sub-second timing while those in the striatum are important for supra-second timing.

## Discussion

Our exploration of neuronal signals in the striatum and the cerebellar dentate nucleus during self-timing yielded four major findings. First, neurons in the striatum were active throughout the delay period and altered the rate of rise of firing rate depending on the timing instruction, while those in the cerebellum were recruited only during the last part of the delay period, exhibiting a similar time course of activity for different interval timings (*Figures 3* and *4*). Second, neuronal correlates of trial-by-trial variation started earlier in the cerebellum than the striatum (*Figure 5*), suggesting that the cerebellum might be essential for the fine adjustment of self-timing in each condition. Third, a subset of striatal neurons showed a clear preference for a specific interval, while such neuron was almost absent in the cerebellum (*Figure 6*). Finally, the effects of inactivation in the striatum were greater for supra-second timing, while those in the cerebellum were dominant for sub-second timing (*Figure 7*), indicating that the relative contributions of these subcortical structures were different for different intervals. Based on these observations, we conclude that neurons in the striatum keep track of elapsed time by representing relative timing to the intended interval, while those in the cerebellum may participate in the generation of self-timed movements with a certain precision.

### Roles of the basal ganglia and cerebellum in self-timing

We found that individual neurons in the striatum continuously kept track of relative timing. A similar scalable ramping activity has been reported in the prefrontal cortex of rats performing the self-timing task within the range of a few seconds (*Xu et al., 2014*). Other recent studies have also shown that the transient neuronal activity in the rat striatum represents a scalable population code for interval timing (*Mello et al., 2015*; *Wang et al., 2018*). Such temporally-dependent transient signals might be integrated in time to generate monotonic ramping activity to keep track of elapsed time and for making decisions (*Janssen and Shadlen, 2005*; *Mita et al., 2009*; *Murakami et al., 2014*). Indeed, some striatal neurons in the latter study were active throughout the delay period and continuously represented passage of time, although the proportion of such neurons appeared to be relatively small (*Mello et al., 2015*).

It has been widely accepted that dopamine signaling in the striatum is essential for interval timing (*Coull et al., 2011*; *Merchant et al., 2013a*). Subjects with Parkinson's disease, for example, show deficits in the estimation and production of interval timing within the range of seconds (*Pastor et al., 1992*) and exhibit a significant decrease in the magnitude of CNV (*Ikeda et al., 1997*). In experimental animals, local application of dopamine receptor antagonists into the striatum alters self-timing in monkeys (*Kunimatsu and Tanaka, 2016*), and more specifically, optogenetic inhibition of nigro-striatal pathways disrupts temporal discrimination in rodents (*Soares et al., 2016*). However, the firing of midbrain dopamine neurons is known to be generally phasic (*Bromberg-Martin et al., 2010*) and therefore appears unlikely to continuously keep track of elapsed time

during motor preparation, while the gain of transient sensory response might carry temporal information (*Soares et al., 2016*). In contrast, the dopamine concentration in the striatum has been recently shown to exhibit a gradual increase when performing tasks requiring continuous behavioral control (*Howe et al., 2013*). Interestingly, in rats moving towards distant goals, the level of striatal dopamine was proportional to the relative distance (or time) to the goals, thereby showing a scalable monotonous increase. Although how the dopamine release is regulated within the striatum remains controversial (*D'Souza and Craig, 2006*), the continuous representation of relative timing in striatal neurons might be relevant to the regulation of dopamine release at the terminal.

Aside from scalable, linear representation of interval timing, the existence of non-linear, duration-selective neuronal representation has also been suggested (*Hayashi et al., 2015*; *Merchant and Averbeck, 2017*). Time-dependent firing modulation in individual neurons has been reported during motor preparation (*Murakami et al., 2014*), sensory-guided continuous movements (*Schoppik and Lisberger, 2006*) and the maintenance of spatial working memory (*Constantinidis and Klingberg, 2016*). In the present experiments, neurons in the striatum showed greater variation in peak firing rate for different mandatory delay intervals than the cerebellum, and a minority of striatal neurons exhibited a clear duration tuning (*Figure 6*). However, in the population as a whole, neuronal activity just before saccade initiation was comparable across interval timing, and the neuronal firing rate at a given moment represented the relative timing to the intended interval. Assuming that the duration-selective elements represent absolute timing, such non-linear temporal representation may not conform to the flexible representation of relative timing found in the neuronal population. Instead, the duration-selective elements might distinguish behavioral goals rather than provide a neuronal representation of elapsed time, although how these signals are used for self-timing remains uncertain.

Since the cerebellum is essential for accurate movements, the generation of self-initiated saccades at proper timing must be one of critical functions of the cerebellum. In this study, we found that neuronal correlates of stochastic variation of self-timed saccade started earlier in the cerebellum than the striatum, while preparatory activity started late in the cerebellum for supra-second intervals (*Figure 5*). Our recent analysis of neuronal activity in the cerebellum showed that trial-by-trial latency for sub-second timing was correlated with the rate of rise of preparatory activity, while that for supra-second timing was correlated with the times of activity onset (*Ohmae et al., 2017*). These results suggest that saccade latency for sub-second timing might depend largely on the magnitude of neuronal activity in the cerebellum, while that for supra-second timing might be regulated by the combination of cerebellar activity and the external signals triggering the activity.

Consistent with this hypothesis, the effects of cerebellar inactivation on saccade latency were dominant for sub-second intervals (*Figure 7D*), while inactivation often increased the variation of self-timing even for supra-second interval (*Figure 7F*). The latter findings were consistent with the previous results of stimulation experiments showing that signals in the cerebellum can modify self-timing for all intervals (*Ohmae et al., 2017*), although the inactivation effects on latency variation found in this study were relatively small for the long intervals. This could be because the cerebellum may have a potential to adjust timing only in the range of several hundreds of milliseconds, and because the inactivation effects of the cerebellum might be masked by the greater variation of duration estimation for supra- than sub-second intervals.

In addition to the increased variation of self-timing, cerebellar inactivation also prolonged saccade latency in the standard memory-guided saccade task and the visually-guided saccade task (*Figure 7D*). Because the cerebellum is known to assist the cerebral cortex to boost motor commands during the initiation of somatic movements (*Thach et al., 1992*), the weak transient activity around the time of visually-triggered saccades found in this study (*Figure 3—figure supplement 3B*) might also play such a role. However, given the strong preparatory activity for self-timing, neurons in our recording sites in the dentate nucleus may not be simply providing motor commands for saccades. While the latency of visually-triggered saccades was only ~180 ms in our monkeys, the ramping activity started ~500 ms before self-timed saccades (*Figure 3—figure supplement 1B*), suggesting a role for motor planning. This was also supported by our previous findings that electrical stimulation applied to the dentate nucleus advanced self-timing without directly eliciting immediate saccades, and that the same stimulation pulses delivered during the delay period did not alter saccade latency in the standard memory-guided saccade task (*Ohmae et al., 2017*). Taken together, these results suggest that the signals in the cerebellar dentate nucleus may regulate timing of decisions for self-initiated movements. This function might gain importance in the situation where the

precision of self-timing need to be preserved. Future studies may require consideration on these possibilities.

As a limitation of the present study, it should be noted that the other parts of the striatum and the deep cerebellar nuclei might represent sub- and supra-second intervals, respectively. Indeed, functional imaging studies often detect multiple loci in respective subcortical structures relevant to temporal information processing (e.g. *Rao et al., 2001*; *Xu et al., 2006*). However, the functional preference for short and long intervals for the cerebellum and the basal ganglia are also supported by many previous functional imaging and case studies (*Lewis and Miall, 2003*; *Ivry and Spencer, 2004*; *Buhusi and Meck, 2005*; *Allman et al., 2014*).

We recently found in monkeys that the trial-by-trial latency of self-timed saccades was inversely correlated with the pupil diameter just before the delay period (*Suzuki et al., 2016*). Importantly, the pupil diameter did not predict absolute timing, but was indicative of relative timing to the intended interval in each trial. Considering that the pupil diameter is well correlated with the noradrenergic signaling in the locus coeruleus (*Aston-Jones and Cohen, 2005*), and that these neurons send massive projections to the cerebellum (*Olson and Fuxe, 1971*), the stochastic variation of self-timing, cerebellar neuronal activity, and the pupil diameter might all be connected. Local manipulation of noradrenergic signaling in the cerebellum can provide a critical test for this possibility, whereas systemic pharmacological application complicates the interpretation of results (*Suzuki and Tanaka, 2017*).

## Integration of subcortical signals

It remains unclear how the signals in the basal ganglia and the cerebellum are integrated to decide movement timing, although these areas have a common connection with the frontal and parietal cortices via the thalamus (*Strick et al., 2009*; *McFarland and Haber, 2001*; *Prevosto et al., 2010*). Recent anatomical data in rodents show that signals from the basal ganglia are sent to the superficial layers in the motor cortex via the thalamus, while those from the cerebellum are sent to deeper layers, suggesting that the basal ganglia and the cerebellum may play roles in preparing and triggering movements, respectively (*Kaneko, 2013*). Similarly, different subcortical signals for self-timing are possibly integrated through interplay between different layers in the cerebral cortex. In fact, individual neurons in the medial frontal cortex in monkeys are known to exhibit different time courses of activity during isochronous tapping (*Merchant and Averbeck, 2017*), suggesting that these signals might come from different subcortical sources. Alternatively, signals originating in the basal ganglia and the cerebellum might be separately processed in different areas in the cerebral cortex. For example, a recent study in rodents demonstrated that neurons in the medial prefrontal cortex signaled deterministic timing, while those in the premotor cortex represented stochastic timing during tasks requiring self-initiated movements (*Murakami et al., 2017*). These cortical areas might be specifically involved in the cortico-basal ganglia and the cortico-cerebellar loops, respectively. Furthermore, recent anatomical studies showed that the basal ganglia and cerebellum can mutually communicate with each other through disynaptic subcortical pathways (*Bostan and Strick, 2018*). Although the outputs from the cerebellar dentate nucleus have been shown to facilitate signals in the striatum through the thalamus in behaving animals (*Chen et al., 2014*), how these subcortical pathways can integrate signals for self-timing need to be examined in future study.

In addition to measuring single time intervals, the functional linkage between the basal ganglia and the cerebellum also appears to be essential for a variety of cognitive tasks. For temporal information processing, beat-based timing has been thought to be processed in the cortico-basal ganglia pathways (*Teki et al., 2011*), but recent analyses also suggest a role for the cerebellum (*Ohmae et al., 2013*; *Teki and Griffiths, 2016*). In addition, while proactive inhibition for demanding behavioral tasks requires intact basal ganglia (*Frank et al., 2007*), recent studies also show that the cerebellum is involved in countermanding (*Ide and Li, 2011*) as well as anti-saccade (*Peterburs et al., 2015*; *Kunimatsu et al., 2016*) paradigms. Furthermore, recent studies in rodents have detected neuronal modulation associated with temporal-difference prediction error of aversive stimuli in the cerebellum (*Ohmae and Medina, 2015*), although such signals have long been thought to be a hallmark of neural processes in the basal ganglia. All these recent studies indicate the necessity of future research linking neuronal processes in the basal ganglia with those in the cerebellum for comprehensive understanding of global network controlling behaviors.

## Materials and methods

### Animal preparation

Experiments were conducted on two female Japanese monkeys (*Macaca fuscata*, 7–8 kg). All experimental protocols were approved by the Hokkaido University Animal Care and Use Committee. Details of surgical procedures for implanting the head holder, eye coil, and recording cylinder are described elsewhere (*Kunimatsu and Tanaka, 2016*). All surgeries were performed using sterile procedures under general isoflurane anesthesia. Analgesics were administered during each surgery and the following few days.

### Behavioral paradigms

During training and experimental sessions, monkeys were seated in a primate chair placed in a darkened booth. Visual stimuli were presented on a 24-inch cathode-ray tube monitor (refresh rate: 60 Hz) that was located 38 cm away from the eyes, and subtended a visual angle of 64 × 44°. We used three saccade paradigms: the self-timed memory-guided saccade task (*Figure 1A*), the standard memory-guided saccade task, and the visually-guided saccade task (*Figure 3—figure supplement 3A*). In the self-timed memory-guided saccade task (*Ashmore and Sommer, 2013*; *Costello et al., 2016*; *Kunimatsu and Tanaka, 2012, 2016*; *Tanaka, 2006, 2007*; *Wang et al., 2018*), monkeys were required to make a saccade to the location of a previously presented visual cue (100 ms) without any immediate external trigger. The FP disappeared only after the animals generated a self-timed saccade, as eye position deviated >3° from the FP. The mandatory delay interval following the cue onset was selected from short (400–700 ms), medium (1000–1600 ms), or long (2400–3100 ms) intervals that were indicated by FP color (*Figure 1A*, lower panel; cyan, blue and yellow squares for short, medium, and long intervals, respectively). In the standard memory-guided saccade task (*Hikosaka and Wurtz, 1983*), monkeys made a saccade to the remembered location of the visual cue in response to the FP offset (<400 ms) that occurred randomly during 700–2500 ms (uniform distribution) following cue onset. In this task, saccade timing was instructed externally by extinction of the FP. In the visually-guided saccade task, the saccade target appeared at the time of the FP offset (always 1600 ms following its appearance), and the animals made an immediate saccade to the target. To inform the monkeys of the trial type, the FP was red for the standard memory-guided saccade task and the visually-guided saccade task (*Figure 3—figure supplement 3A*). In all these tasks, the saccade target and visual cue were presented either 16° left or right of the FP. The size of the eye position window was 2° for initial fixation and 4° for the peripheral targets. Correct performance was reinforced with a liquid reward at the end of each trial. Each trial was presented in pseudo-random order within a block that consisted of 10 different trials (five trials in opposite directions).

### Recording and inactivation procedures

To record from single neurons, a glass-coated tungsten electrode (Alpha Omega Engineering) guided by a 23-gauge stainless tube was lowered into the striatum (caudate nucleus, ±2 mm of the anterior commissure and 4–8 mm lateral to the midline) or the cerebellar dentate nucleus (6–8 mm posterior to the interaural line and 6–8 mm lateral) in separate experiments using a micromanipulator (MO-97S; Narishige). The location of electrode penetration was adjusted using the grid system (Crist Instruments). Signals obtained from the electrodes were amplified and filtered (0.3–10 kHz, Model 1800; A-M Systems). The waveform of action potentials of a single neuron was isolated using a real-time spike sorter with template-matching algorithms (ASD; Alpha Omega Engineering). We searched for the task-related neurons when monkeys performed a block of randomized saccade trials. Once we isolated the putative task-related neuron exhibiting a ramping activity before saccade initiation, we collected data for ≥6 trials for the further offline analysis. We did not include the data from presumably tonically active neurons in the caudate nucleus, which exhibited characteristic tonic firing pattern and wider action potentials (*Aosaki et al., 1995*). Neurons included for the analysis had low baseline firing rate and were considered as medium-spiny projection neurons and some GABAergic interneurons.

For the inactivation experiments, we manufactured injectrodes composed of epoxy-coated tungsten microelectrode (FHC Inc.) and silica tube (Polymicro Technology; *Tachibana et al., 2008*). The injectrode connected to a 10 μL Hamilton microsyringe was inserted through the guide tube, and a

small amount of GABA$_A$ agonist (muscimol; 5 µg/µL, 1 µL for each site) was pressure-injected using a micropump (NanoJet; Chemyx Inc.) at the sites where the task-related neurons were previously recorded. For the inactivation experiments in the striatum, we infused muscimol into two sites along the same penetration (>500 µm apart). The inactivation effects were assessed by comparing eye movements before and 15–90 min after muscimol injection. We also injected saline in separate experiments to ensure that the effect was not due to any volume effect.

## Histological procedures

Recording sites in one monkey were reconstructed from histological sections (*Figure 2*). At the end of the experiments, several electrolytic lesions were made at or near sites where task-related neurons were recorded by passing a direct current through the electrodes (10–20 µA, tip negative, 800–1000 µC). The animal was then deeply anesthetized with sodium pentobarbital (>60 mg/kg, i.p.) and perfused transcardially with 0.1 M phosphate buffer followed by 3.5% formalin. The brain was cut into 50 µm thick transverse (cerebellum) and coronal (striatum) sections using a freezing microtome, and each section was stained with cresyl violet. We reconstructed the location of each task-related neuron according to the depth and coordinates of electrode penetrations, and the locations relative to the marking lesions.

## Data acquisition and analysis

Spike timing and eye movement data were sampled at 1 kHz and saved in files during the experiments. Further off-line analyses were performed using Matlab (MathWorks). Saccades were detected when angular eye velocity exceeded 40°/s and eye displacement was >3°. Trials were excluded from the analysis when saccades landed >5° from the cue location. Saccade latency was defined as the time from either cue onset (self-timed saccade task) or FP offset (other two tasks) to the time of saccade initiation. Although monkeys sometimes generated early saccades before the expiration of mandatory delay for reward (6–12% and 16–35% of self-timed trials for monkeys B and G, respectively, *Figure 1B*), these trials were also included for the analysis because the animals were supposed to monitor the elapsed time.

For quantification, we measured the firing rate during the following periods: (1) 300 ms just before the fixation point onset (baseline period), (2) 200 ms after the cue onset (visual period), (3) the 350 ms interval starting from 500 ms before saccades (delay period), and (4) 150 ms before saccade initiation (saccade period). In this study, we considered neurons with a significant firing modulation during the delay period according to post hoc multiple comparisons (one-way ANOVA followed by Scheffé's method, p<0.05). The time course of neuronal activity for each condition was qualitatively examined by computing the spike density function using a Gaussian kernel (σ = 15 ms).

To compare the time courses of preparatory activity for neurons in different structures, we estimated the rate of rise of neuronal activity by fitting (least squares) a non-linear function with parameters of the baseline, slope, and onset time of ramp-up activity (*Figure 4A*). The normalized population activity was computed from the traces of spike density starting from 400 ms before cue onset and ending at the mean saccade latency. The data during the 200 ms following 50 ms after cue onset were excluded to eliminate visual transient. To obtain confidence intervals, we randomly resampled neurons with replacement to obtain replications of the same size as the original data set, and this was repeated 1000 times (bootstrap method, *Figure 4B and C*).

For the analysis of neuronal correlates of trial-by-trial variation in self-timing, the data for each neuron were divided into three groups according to saccade latencies, and then the population of normalized activity for saccades in the preferred direction was computed for each group (*Figure 5*). We compared normalized activity among three groups every millisecond (repeated measures ANOVA) and measured the timing of neuronal variation relative to either cue onset or saccade initiation when the traces significantly diverged (p<0.01) for >40 ms (*Figure 5C*). The distributions of timing of neuronal variation were again computed using the bootstrap method.

For the triangular plots in *Figure 6B*, the firing rate before saccades (200 ms) for each neuron was normalized for the sum of the activity in different conditions. The x-coordinate of each data point is the difference in normalized activity between the medium and long delay intervals, and the y-coordinate is the normalized response in the short delay interval times $\sqrt{3}$.

## Acknowledgements

We thank T Mori, A Hironaka for technical assistance, M Suzuki for her administrative help, M Takei and Y Hirata for manufacturing equipment, and all lab members for comments and discussions. We are also grateful to M Takada and K Inoue in the Primate Research Institute of Kyoto University for their insightful comments regarding the recording sites. Animals were provided by the National Bio-Resource Project. This work was supported partly by grants from the Ministry of Education, Culture, Sports, Science and Technology of Japan, the Takeda Science Foundation, and the Uehara Memorial Foundation.

## Additional information

### Funding

| Funder | Grant reference number | Author |
|---|---|---|
| Ministry of Education, Culture, Sports, Science and Technology | 24800001 | Jun Kunimatsu |
| Ministry of Education, Culture, Sports, Science and Technology | 25119005 | Masaki Tanaka |
| Ministry of Education, Culture, Sports, Science and Technology | 17H03539 | Masaki Tanaka |
| Ministry of Education, Culture, Sports, Science and Technology | 18H05523 | Masaki Tanaka |
| Takeda Science Foundation | | Masaki Tanaka |
| Uehara Memorial Foundation | | Masaki Tanaka |

The funders had no role in study design, data collection and interpretation, or the decision to submit the work for publication.

### Author contributions

Jun Kunimatsu, Conceptualization, Data curation, Formal analysis, Validation, Investigation, Visualization, Writing—original draft, Writing—review and editing; Tomoki W Suzuki, Shogo Ohmae, Investigation; Masaki Tanaka, Conceptualization, Supervision, Funding acquisition, Writing—original draft, Project administration, Writing—review and editing

### Author ORCIDs

Jun Kunimatsu http://orcid.org/0000-0002-8003-0650
Tomoki W Suzuki http://orcid.org/0000-0002-7085-1594
Shogo Ohmae http://orcid.org/0000-0003-1726-4961
Masaki Tanaka http://orcid.org/0000-0002-6177-1314

### Ethics

Animal experimentation: All experimental protocols were evaluated and approved by the Hokkaido University Animal Care and Use Committee (#13-0114). All surgery was performed under general isoflurane anesthesia, and every effort was made to minimize suffering.

### Decision letter and Author response

Decision letter https://doi.org/10.7554/eLife.35676.025
Author response https://doi.org/10.7554/eLife.35676.026

## Additional files

**Supplementary files**
• Transparent reporting form
DOI: https://doi.org/10.7554/eLife.35676.023

**Data availability**
Numerical data for main figures and figure supplements have been provided as source data files.

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
