## [Decision Letter]

Thank you for submitting your article "Different contributions of preparatory activity in the basal ganglia and cerebellum to self-timing" for consideration by *eLife*. Your article has been reviewed by three peer reviewers, including Naoshige Uchida as the Reviewing Editor and Reviewer #1, and the evaluation has been overseen by Richard Ivry as the Senior Editor. The following individuals involved in review of your submission have agreed to reveal their identity: Robert S Turner (Reviewer #2); John Assad (Reviewer #3).

The reviewers have discussed the reviews with one another and the Reviewing Editor has drafted this decision to help you prepare a revised submission.

The study compares the firing patterns of neurons in the striatum (the anterior part of the caudate nucleus) and the cerebellum (the posterior part of the cerebellar dentate nucleus) in a self-timed saccade task in two Japanese monkeys. Monkeys were trained to make a saccadic eye movement after a brief target cue presentation (100ms). To obtain reward, saccade onset had to be within a predetermined time interval (short, 400-700ms; medium, 1000-1600ms; long, 2400-3100ms) signaled by the color of the fixation point. The authors recorded the activity of neurons that 'elevated activity before self-timed saccade'. The authors made several interesting findings. First, the activity in the striatum started ramping shortly after the cue onset while that in the cerebellar nucleus started ramping only shortly before saccadic eye movement. Second, trial-by-trial fluctuation of saccade onset is more correlated with the pattern of activity in the cerebellar nucleus but not with that in the caudate nucleus. Third, the authors performed unilateral transient inactivation of each region using muscimol. During caudate inactivation, the onset of contraversive saccades was delayed in long interval trials, whereas that of ipsiversive saccades was shortened for all intervals. During cerebellar nucleus inactivation, a significant effect was found only for ipsiversive saccades in short interval trials. Furthermore, the coefficient of variation (CV) of saccade onset was not altered by caudate inactivation but increased in some conditions by cerebellar nucleus inactivation.

The possibility that neurons in these regions deferentially regulate self-timed movement initiation is of great interest, and this manuscript contains potentially very interesting results. The study is well designed, and the results appear to be high quality. The analyses are mostly appropriate (but see below) and text and illustrations are well prepared. Although all the three reviewers found this study to be of importance, they raised some concerns that we would like to see your response before recommending this work for publication at *eLife*.

Essential revisions:

1) The analyses of ramping activity were performed by aligning trials to cue onset. However, these analyses alone do not distinguish alternative hypotheses that need to be separated. For instance, these analyses do not distinguish whether the variability in ramping activity comes from different ramping patterns (e.g. different rates in ramp) or merely from the variation in movement onset timing. Some of the interpretations must distinguish these possibilities. It is, therefore, important to analyze the data by aligning trials to movement onset. We appreciate that you have already performed this in some analyses, but it is important to apply this method further and to more carefully interpret the results. Specifically, the measurement of ramping slope (Figure 4) should be made on data aligned to movement onset rather than cue onset. Moreover, a more conservative interpretation of the data (at least in Figure 5) is that the striatum activates considerably earlier than the cerebellum during self-timed movements; the "variability" argument is less convincing. That simpler interpretation is still consistent with the different roles that have been proposed for striatum and cerebellum in timing (seconds vs. sub-second).

Reviewer 3 provided a very thoughtful analysis of this issue, as summarized in the following paragraphs (taken directly from his/her review with some editing).

The major problem is that much of the analysis was performed with responses aligned to CUE onset rather than MOVEMENT onset (Figure 3 and Figure 4; I have seen this also in several other papers claiming that neurons show ramping before self-timed movements). The issue is that there is a *distribution* of movement times following cue onset. Most neurons in the motor system show activity preceding movement; these responses can be highly stereotyped relative to movement onset. If trials are aligned to the *cue time*, then on average among trials, even a stereotyped movement-related response will appear as a "ramp", because the variability in self-timed movement latency from trial-to-trial will smear out the average response. Even more dangerous, the distribution of movement times is broader with longer timed intervals than shorter intervals, as the authors showed (Figure 1B) and many other labs have reported. Thus, if trials are averaged aligned on cue onset, the "ramp" will appear shallower for late movements than early movements – i.e., the broader distribution of movement times will "smear out" the average activity more for long than short intervals. This is exactly the problem in Figure 3 and Figure 4. These data need to be re-analyzed aligned with respect to *movement time*, not cue time (i.e., averaging trials back from the movement onset, excluding the 200 ms period following the cue to omit the visual response to the cue). The cue-aligned smearing problem is especially a problem for the cerebellar data, because the apparent "ramp" begins only ~500 ms before the movement (as the authors note) and thus overlaps extensively with the distribution of movement times.

This has been done in Figure 5. Here the average striatal responses (Figure 5A) still show clear differences in ramping slopes between short, medium and long times, but there is not much difference in slope evident for the cerebellar neurons (Figure 5B). That is, the *slopes* of the rise in cerebellar activity do not appear so different between the sub-divided trials in each category (e.g., light blue, medium blue and dark blue), or between medium or long trials (short trials are difficult to compare, because they are likely contaminated with the cue response). The ramp slope should be quantitatively measured from these movement-aligned data, not from cue-aligned data.

These observations make it hard to evaluate the claim that the ramps diverge earlier in cerebellum than in striatum. Figure 5A may suggest that striatal activity before the movement is actually comprised of two components, a slow ramp with a slope that co-varies with movement time (short, medium or long) and fast, peri-movement response that is more stereotyped with respect to the movement onset. The cerebellar neurons seem to only have only a fast, peri-movement response, which is larger than the peri-movement response of the striatal neurons. The cerebellar neurons also have much higher intrinsic firing rates. This difference may make it harder to detect the divergence point of the striatal peri-movement component compared to the cerebellum. It is also problematic to argue that the cerebellar response controls the late-phase variability of the movement latency when the cerebellar responses appears to be so stereotyped relative to movement. That is, the stereotypy of the cerebellar response might suggest it arises "after the decision" to move whereas the more variable slow ramp in striatum might suggest it "contributes to the decision". As written now, casual readers may walk away with the idea that the cerebellum is somehow injecting variability into the movement time, whereas cerebellar responses seems fairly stereotyped relative to movement time.

2) Interpretation and General conclusion: The general conclusion regarding the role of the cerebellum hardly reaches beyond the results and thus is relatively uninformative. The authors conclude that Cb "might be primarily responsible for the stochastic variation of self-timing." This, of course, is just what the data show. But why would such a large and complicated structure as the Cb have such a simple job as adding noise to interval timing? Is there some advantage to the animal to implement "stochastic variation of self-timing?" In other words, is sub-second variation an aspect of behavior that benefits the animal and that the Cb is designed (by evolution) to implement? Or, do the authors believe that this kind of variation is merely performance noise? If performance noise, then is it possible the Cb is reflecting this variability without actually controlling it?

It appears that another way to describe the same results is that activity in the Cb is closely time-locked to the time of saccade onset, even that Cb is closely involved in triggering saccade onset. Granted, the Discussion section does allude to ideas that Cb may be involved in action triggering, but this explanation differs significantly from the one invoked throughout the manuscript ascribing a role to the Cb in "variation control".

3) Please provide more information regarding the rationale and the details of recording conditions. First, please provide more information concerning the locations sampled in striatum and dentate. Why were these locations chosen? What are the stereotaxic locations of the regions sampled? Were identical locations sampled in both animals? Do these regions of striatum and dentate "project to" (via their trans-thalamic pathways) the same regions of cortex? Is it possible that a different part of striatum might show sub-second variation related activity? or a different part of dentate show multi-second ramping activity?

A related question is whether the authors discriminated between the identifiable neuronal subtypes in the striatum? Were all or most of the neurons likely spiny projection neurons? Were TANs explicitly excluded from analysis?

4) To analyze the onset and slope of ramping activities, the authors fit a line incorporating onset and baseline (three parameters). It would be important to discuss how good the fit is (goodness of fit) at the level of single neurons and populations. Furthermore, it is important to report some data regarding neuron-to-neuron variability in firing patterns.

5) The interpretation of the inactivation experiments appears to warrant further clarification. There appears to be a significant change in the CV in the long interval condition (Figure 7F). Is this true? How does this relate to the authors' conclusions? Overall, it would be helpful to clarify how the authors interpret the inactivation results (Figure 7).

6) Last, but not least, there is a logical concern with your study. You compare activity in the input nucleus of the basal ganglia with activity in an output nucleus of the cerebellum. This difference is especially relevant given that current computational models of timing in these subcortical structures tend to focus on the striatum and cerebellar cortex (either Purkinje cells or the cerebellar cortical cells that provide input to the Purkinjes). We are not aware of models that posit a role for DCN neurons in "timing computations". In fact, some classic results in the eyeblink literature have shown that if the system is just driven by DCN (following lesions/inactivation of cerebellar cortex), you lose the adaptive timing of the CR. This has been taken to mean that inputs to DCN (from extracerebellar regions) are sufficient to associate the CS and US, but the adaptive timing of the CR comes from the cerebellar cortex.

This issue needs to be made explicit both in the Introduction and Discussion section. You need to outline the logic of comparing the striatum to the DCN, as well as outline how your results may be qualified by the fact that you have focused on this comparison. As things are now written, the framing makes it sound like the striatum and DCN provide a way to compare temporal processing/representation in the basal ganglia and cerebellum. For example, in the Introduction, you write, "These results indicate that neurons outside of the cerebellum must keep track of elapsed time to make temporally accurate movements in trials with suprasecond delay intervals", and then go on to make the argument that for turning to the striatum. This is reasonable, but note you are going from DCN observations to the conclusion that the tracking of elapsed time must be from "neurons outside of the cerebellum." This ignores the possibility that the tracking might be done in the cerebellar cortex with the projection from the cerebellar cortex to DCN introducing some sort of non-linearity/thresholding function that obscures more of a ramping-like function.

[Editors' note: further revisions were requested prior to acceptance, as described below.]

Thank you for resubmitting your work entitled "Different contributions of preparatory activity in the basal ganglia and cerebellum for self-timing" for further consideration at *eLife*. Your revised article has reviewed by two peer reviewers, including Nao Uchida as the Reviewing Editor and Reviewer #1, and the evaluation has been overseen by Rich Ivry as the Senior Editor.

The manuscript has been improved but there is one final issue that we would like to see addressed before making a final decision. Specifically, the reviewers thought that the data do not provide strong support for the broad claim that 'the cerebellum might be primarily responsible for the stochastic variation of self-timing'. We ask that you qualify this claim/inference in the manuscript.

I'm passing along the comments from reviewer 3 on this issue to help guide you in this endeavor.

Reviewer #3:

The paper has been improved. However, I am still skeptical of the claim that the cerebellum is more responsible for stochastic variation in self-timing than the striatum. The authors have tried to address this issue, but the data/arguments are not convincing. For example, in the Discussion section, the authors suggest that the cerebellum adds or maintains variation of saccade timing – but only for short estimated intervals. For longer intervals, the authors state "there was no need for the animals to maintain the variation of saccade timing within several hundreds of milliseconds". This begs the question of why there is there a "need" at all to maintain variation in timing. Perhaps inherent variation allows us to explore different intervals in a world with constantly changing temporal contingencies - but if so, why would we relax that exploratory bent for supra-second intervals?

Stepping back, there were two observations behind the claim that the cerebellum is responsible for stochastic variation. First, for the medium and long intervals, if trials are grouped according to precise movement time, the neuronal responses in the Cb appear to diverge a few hundred ms earlier than those of the striatum (Figure 5). This earlier divergence is based (per force) on a statistical criterion – when significant differences first arise based on the results of repeated-measures ANOVA. This is not a fair comparison between caudate and Cb. Average firing rates in caudate are far lower than in Cb dentate firing (Figure 3C-D). Moreover, I'm not sure about dentate nucleus firing statistics, but firing patterns in the Cb can be highly regular (e.g., Purkinje cells), which would reduce variance in spike counts for a given mean firing rate. For both these reasons, it might just be easier to DETECT the ANOVA-based divergence point for the cerebellar neurons than the caudate neurons, whereas we don't know how the brain actually pools signals from these two areas. I'm not exactly sure how to address this problem, but the authors could try to repeat their analysis with a subset of data matched for mean firing rates from the two areas (if possible). It would also be helpful to at least examine the coefficient of variation of firing rates between the two areas.

Second, the authors conclude the description of the inactivation experiment (end of Results section) with the statement that stochastic variation "might be controlled by signals in the cerebellum, but not in the striatum" - but this does not seem warranted from the inactivation experiment. The clearest result is that inactivation of Cb affected mean movement time most for short intervals while striatum affected mean movement time most for medium and long times. With respect to *variation* in latency, the effects of Cb inactivation were inconsistent at best. In addition, the largest change in movement-time variation with Cb inactivation was for short intervals, but this also caused the largest shift in mean movement latency of any area/interval/experiment, which might be expected to automatically increase the variation in latency (e.g., as per scalar expectancy theory of interval timing). But more generally, what is the hypothesized effect of inactivation on latency variation? Do the authors posit that the cerebellum controls precision, and thus that Cb inactivation would erode that precision? Is that interpretation not at odds with the results in Figure 5? Or do the authors hypothesize that the Cb adds noise to the timing process, in which case Cb inactivation might reduce variation?

In the end, the clearest result in the paper is the differential roles for the caudate and Cb for long and short intervals, respectively, supported by both electrophysiological recording and inactivation experiments. This alone is a nice contribution. The stochastic variation claims are not strongly supported (either empirically or theoretically). Of course, the authors could present the "divergence" argument in Figure 5, but in a more conservative manner, pointing out the difficulty in statistically distinguishing divergence points between neuronal populations with different firing rates and potentially different firing statistics. But I don't think that the cerebellar stochastic variation claims should be featured prominently in the paper (for example, the abstract should not conclude with that claim).

We also ask that you double check the new supplement 1 to Figure 3 (saccade-aligned responses). The SD of cue times seems narrow given the wide distributions of movement times in Figure 1B. Please verify that the figure is correct.

---

## [Author Response]

Essential revisions:1) The analyses of ramping activity were performed by aligning trials to cue onset. However, these analyses alone do not distinguish alternative hypotheses that need to be separated. For instance, these analyses do not distinguish whether the variability in ramping activity comes from different ramping patterns (e.g. different rates in ramp) or merely from the variation in movement onset timing. Some of the interpretations must distinguish these possibilities. It is, therefore, important to analyze the data by aligning trials to movement onset. We appreciate that you have already performed this in some analyses, but it is important to apply this method further and to more carefully interpret the results. Specifically, the measurement of ramping slope (Figure 4) should be made on data aligned to movement onset rather than cue onset. Moreover, a more conservative interpretation of the data (at least in Figure 5) is that the striatum activates considerably earlier than the cerebellum during self-timed movements; the "variability" argument is less convincing. That simpler interpretation is still consistent with the different roles that have been proposed for striatum and cerebellum in timing (seconds vs. sub-second).Reviewer 3 provided a very thoughtful analysis of this issue, as summarized in the following paragraphs (taken directly from his/her review with some editing).The major problem is that much of the analysis was performed with responses aligned to CUE onset rather than MOVEMENT onset (Figure 3 anf Figure 4; I have seen this also in several other papers claiming that neurons show ramping before self-timed movements). The issue is that there is a *distribution* of movement times following cue onset. Most neurons in the motor system show activity preceding movement; these responses can be highly stereotyped relative to movement onset. If trials are aligned to the *cue time*, then on average among trials, even a stereotyped movement-related response will appear as a "ramp", because the variability in self-timed movement latency from trial-to-trial will smear out the average response. Even more dangerous, the distribution of movement times is broader with longer timed intervals than shorter intervals, as the authors showed (Figure 1B) and many other labs have reported. Thus, if trials are averaged aligned on cue onset, the "ramp" will appear shallower for late movements than early movements – i.e., the broader distribution of movement times will "smear out" the average activity more for long than short intervals. This is exactly the problem in Figure 3 and Figure 4. These data need to be re-analyzed aligned with respect to *movement time*, not cue time (i.e., averaging trials back from the movement onset, excluding the 200 ms period following the cue to omit the visual response to the cue). The cue-aligned smearing problem is especially a problem for the cerebellar data, because the apparent "ramp" begins only ~500 ms before the movement (as the authors note) and thus overlaps extensively with the distribution of movement times.This has been done in Figure 5. Here the average striatal responses (Figure 5A) still show clear differences in ramping slopes between short, medium and long times, but there is not much difference in slope evident for the cerebellar neurons (Figure 5B). That is, the *slopes* of the rise in cerebellar activity do not appear so different between the sub-divided trials in each category (e.g., light blue, medium blue and dark blue), or between medium or long trials (short trials are difficult to compare, because they are likely contaminated with the cue response). The ramp slope should be quantitatively measured from these movement-aligned data, not from cue-aligned data.These observations make it hard to evaluate the claim that the ramps diverge earlier in cerebellum than in striatum. Figure 5A may suggest that striatal activity before the movement is actually comprised of two components, a slow ramp with a slope that co-varies with movement time (short, medium or long) and fast, peri-movement response that is more stereotyped with respect to the movement onset. The cerebellar neurons seem to only have only a fast, peri-movement response, which is larger than the peri-movement response of the striatal neurons. The cerebellar neurons also have much higher intrinsic firing rates. This difference may make it harder to detect the divergence point of the striatal peri-movement component compared to the cerebellum. It is also problematic to argue that the cerebellar response controls the late-phase variability of the movement latency when the cerebellar responses appears to be so stereotyped relative to movement. That is, the stereotypy of the cerebellar response might suggest it arises "after the decision" to move whereas the more variable slow ramp in striatum might suggest it "contributes to the decision". As written now, casual readers may walk away with the idea that the cerebellum is somehow injecting variability into the movement time, whereas cerebellar responses seems fairly stereotyped relative to movement time.

We agree with the reviewer in that the variation of saccade timing must obscure the slope of ramping activity when the data are aligned with the cue onset. In the revised manuscript, we have added the data for the analysis of ramping activity with saccade alignment (Figure 4C). We also present the population data aligned with saccade initiation (Figure 3—figure supplement 1). The results were in good agreement with the original data aligned with the cue onset. The times of ramp onset greatly differed for different interval timing in the striatum but were comparable in the cerebellum (Figure 4C). The degree of changes in ramp slope for different intervals was greater in the striatum than the cerebellum (Figure 4D), although the slope differed for different intervals in both structures. We have modified the text in the Results section accordingly. Also, we have corrected the unit on the ordinate of Figure 4B right panel (from spikes/s/s to unit/s), because the quantification was made for the normalized activity as shown in Figure 4A.

As the reviewer pointed out, the preparatory activity in the cerebellum appears to be more stereotyped than that in the striatum. This was further elucidated as we quantified the inter-neuronal variation in response to the comment #4 below (Figure 3—figure supplement 2). However, we do not think that the cerebellum simply triggers movements following motor decisions, because the ramping activity was found before self-timed saccades only and started well before the movements (~500 ms) while the triggering of saccades occurred within ~180 ms in the standard memory-guided saccade task. Taken together with the results of inactivation and our previous stimulation study performed in other animals (Ohmae et al., 2017), we have added a paragraph to thoroughly discuss this important point (subsection “Roles of the basal ganglia and cerebellum in self-timing”, "In addition to the increased variation of self-timing, cerebellar inactivation also prolonged saccade latency in the standard memory-guided saccade task and the visually-guided saccade task (Figure 7D). […] Taken together, these results suggest that the signals in the cerebellar dentate nucleus may regulate timing of decisions for self-initiated movements. This function might gain importance in the situation where the precision of self-timing need to be preserved. Future studies may require consideration on these possibilities.").

2) Interpretation and General conclusion: The general conclusion regarding the role of the cerebellum hardly reaches beyond the results and thus is relatively uninformative. The authors conclude that Cb "might be primarily responsible for the stochastic variation of self-timing." This, of course, is just what the data show. But why would such a large and complicated structure as the Cb have such a simple job as adding noise to interval timing? Is there some advantage to the animal to implement "stochastic variation of self-timing?" In other words, is sub-second variation an aspect of behavior that benefits the animal and that the Cb is designed (by evolution) to implement? Or, do the authors believe that this kind of variation is merely performance noise? If performance noise, then is it possible the Cb is reflecting this variability without actually controlling it?It appears that another way to describe the same results is that activity in the Cb is closely time-locked to the time of saccade onset, even that Cb is closely involved in triggering saccade onset. Granted, the Discussion section does allude to ideas that Cb may be involved in action triggering, but this explanation differs significantly from the one invoked throughout the manuscript ascribing a role to the Cb in "variation control".

The reviewer is right. Although the results of inactivation experiments clearly show that the signals in the cerebellar dentate nucleus controls the accuracy and precision of self-timing, it is difficult to interpret the results in terms of behavioral benefit in our experimental paradigm, at least for the long delay condition. We have inserted some lines in the Discussion section to mention this point, "This could be because the cerebellum may have a potential to adjust timing only in the range of several hundreds of milliseconds, and because the inactivation effects of the cerebellum might be masked by the greater variation of duration estimation for supra- than sub-second intervals. In other words, while our results suggest that the stochastic variation of self-timing may primarily reflect neuronal signals in the cerebellum, the cerebellum might not be fully engaged in current behavioral condition. For example, when the mandatory delay was long (e.g., 2200 ms), there was no need for the animals to maintain the variation of saccade timing within several hundreds of milliseconds.").

As for the alternative interpretation of triggering of saccades, we have added a paragraph to the Discussion section considering the roles for the cerebellum in self-timing, as stated in the response to the major comment #1 above. Because the preparatory activity started well before movements and was observed only for self-timed saccades, neuronal signals in the dentate nucleus are likely to play a role beyond simply triggering movements.

3) Please provide more information regarding the rationale and the details of recording conditions. First, please provide more information concerning the locations sampled in striatum and dentate. Why were these locations chosen? What are the stereotaxic locations of the regions sampled? Were identical locations sampled in both animals? Do these regions of striatum and dentate "project to" (via their trans-thalamic pathways) the same regions of cortex? Is it possible that a different part of striatum might show sub-second variation related activity? or a different part of dentate show multi-second ramping activity?

We chose the anterior striatum and the posterior dentate nucleus because they commonly project to the areas in the frontal and parietal cortices known to be related to self-timing, and because the previous studies found neurons associated with saccades in respective subcortical regions. Along with these information (Introduction), we now report the stereotaxic coordinates of recording sites in both monkeys(Materials and methods section). The last question is far beyond the scope of the present study, and we are unable to exclude these possibilities unless thoroughly recording from both structures. We have added a paragraph to the Discussion section to mention this ("As a limitation of the present study, it should be noted that the other parts of the striatum and the deep cerebellar nuclei might represent sub- and supra-second intervals, respectively. Indeed, functional imaging studies often detect multiple loci in respective subcortical structures relevant to temporal information processing (e.g., Rao et al., 2001; Xu et al., 2006). However, the functional preference for short and long intervals for the cerebellum and the basal ganglia are also supported by many functional imaging and case studies (Lewis and Miall, 2003; Ivry and Spencer, 2004; Buhusi and Meck, 2005; Allman et al., 2014).").

A related question is whether the authors discriminated between the identifiable neuronal subtypes in the striatum? Were all or most of the neurons likely spiny projection neurons? Were TANs explicitly excluded from analysis?

During experiments, we discriminated neuron type based on the baseline firing rate and the width of action potentials. Technically, we were able to exclude the tonically active neurons (TANs) easily, but our sample might also contain GABAergic interneurons in addition to the projection neurons. We have noted this point in the revised text (subsection “Recording and inactivation procedures”, "We did not include the data from presumably tonically active neurons in the caudate nucleus, which exhibited characteristic tonic firing pattern and wider action potentials (Aosaki et al., 1995). Neurons included for the analysis had low baseline firing rate and were considered as medium-spiny projection neurons and some GABAergic interneurons.").

4) To analyze the onset and slope of ramping activities, the authors fit a line incorporating onset and baseline (three parameters). It would be important to discuss how good the fit is (goodness of fit) at the level of single neurons and populations. Furthermore, it is important to report some data regarding neuron-to-neuron variability in firing patterns.

We now report the distribution and range of coefficient of determination of fitting for the population data in the Results section. As stated in the text, the quantification by line fitting was performed on the population activity obtained from each bootstrap resampling, not on individual neuronal activity. As suggested, we have compared the inter-neuronal variation of the time courses of preparatory activity between the recoding sites (Figure 3—figure supplement 2). As stated in the text (subsection “Time courses of preparatory activity for self-timing in the striatum and the cerebellum”) and in the figure legend, neuronal activity during the delay interval was more variable in the striatum than the cerebellum.

5) The interpretation of the inactivation experiments appears to warrant further clarification. There appears to be a significant change in the CV in the long interval condition (Figure 7F). Is this true? How does this relate to the authors' conclusions? Overall, it would be helpful to clarify how the authors interpret the inactivation results (Figure 7).

The effects of inactivation on saccade latency were verified by Wilcoxon rank-sum test with Bonferroni correction for individual experiments and by paired t-test for multiple experiments. We also verified the effects of inactivation on the variance of saccade latency by F-test with Bonferroni correction. These are stated in the Results section of the revised text. As stated in the Results section, we found significant changes in standard deviation (SD) for short and long delay intervals (Figure 7F), while changes in coefficient of variation (CV) were found only for short delay interval. Along with the discussion regarding the weak inactivation effects on latency variation for supra-second intervals, we have inserted some lines explaining the possible limitation of the experimental paradigm, as explained above in the response to the comment #2 (subsection “Roles of the basal ganglia and cerebellum in self-timing”, "…although the inactivation effects on latency variation found in this study were relatively small for the long intervals. This could be because the cerebellum may have a potential to adjust timing only in the range of several hundreds of milliseconds, and because the inactivation effects of the cerebellum might be masked by the greater variation of duration estimation for supra- than sub-second intervals. In other words, while our results suggest that the stochastic variation of self-timing may primarily reflect neuronal signals in the cerebellum, the cerebellum might not be fully engaged in current behavioral condition. For example, when the mandatory delay was long (e.g., 2200 ms), there was no need for the animals to maintain the variation of saccade timing within several hundreds of milliseconds.").

6) Last, but not least, there is a logical concern with your study. You compare activity in the input nucleus of the basal ganglia with activity in an output nucleus of the cerebellum. This difference is especially relevant given that current computational models of timing in these subcortical structures tend to focus on the striatum and cerebellar cortex (either Purkinje cells or the cerebellar cortical cells that provide input to the Purkinjes). We are not aware of models that posit a role for DCN neurons in "timing computations". In fact, some classic results in the eyeblink literature have shown that if the system is just driven by DCN (following lesions/inactivation of cerebellar cortex), you lose the adaptive timing of the CR. This has been taken to mean that inputs to DCN (from extracerebellar regions) are sufficient to associate the CS and US, but the adaptive timing of the CR comes from the cerebellar cortex.This issue needs to be made explicit both in the Introduction and Discussion section. You need to outline the logic of comparing the striatum to the DCN, as well as outline how your results may be qualified by the fact that you have focused on this comparison. As things are now written, the framing makes it sound like the striatum and DCN provide a way to compare temporal processing/representation in the basal ganglia and cerebellum. For example, in the Introduction, you write, "These results indicate that neurons outside of the cerebellum must keep track of elapsed time to make temporally accurate movements in trials with suprasecond delay intervals", and then go on to make the argument that for turning to the striatum. This is reasonable, but note you are going from DCN observations to the conclusion that the tracking of elapsed time must be from "neurons outside of the cerebellum." This ignores the possibility that the tracking might be done in the cerebellar cortex with the projection from the cerebellar cortex to DCN introducing some sort of non-linearity/thresholding function that obscures more of a ramping-like function.

As the reviewer pointed out, the previous studies of eye blink conditioning clearly show that the cerebellar cortex is necessary for learning of stimulus timing while the deep cerebellar nuclei (DCN) only are sufficient for the expression of conditioned response. Furthermore, it appears to be reasonable that the local circuits within the cortex play a major computational role in the cerebellum. However, because the DCN (and the vestibular nuclei) serve as only output node of the cerebellum, any results of computation in the cerebellum must be transmitted through the DCN to the other brain regions. We therefore recorded from the dentate nucleus to explore the neuronal representation of timing in the cerebellum. To clarify this, we have added a few sentences to the revised Introduction ("Although the previous studies of eye blink conditioning suggest that plasticity in the cerebellar cortex rather than the deep cerebellar nuclei plays a role in the learning of motor timing (Garcia and Mauk, 1998; Perrett et al., 1993; Raymond et al., 1998), neurons in the nuclei encode timing signals originated in the cerebellar cortex (Ten Brinke et al., 2017). Therefore, we explored signals in the dentate nucleus in this study because any significant computation in the lateral cerebellum must modify neuronal activity in the nucleus that may eventually regulate movement timing.").

[Editors' note: further revisions were requested prior to acceptance, as described below.]

The manuscript has been improved but there is one final issue that we would like to see addressed before making a final decision. Specifically, the reviewers thought that the data do not provide strong support for the broad claim that 'the cerebellum might be primarily responsible for the stochastic variation of self-timing'. We ask that you qualify this claim/inference in the manuscript.I'm passing along the comments from reviewer 3 on this issue to help guide you in this endeavor.Reviewer #3:The paper has been improved. However, I am still skeptical of the claim that the cerebellum is more responsible for stochastic variation in self-timing than the striatum. The authors have tried to address this issue, but the data/arguments are not convincing. For example, in the Discussion section, the authors suggest that the cerebellum adds or maintains variation of saccade timing – but only for short estimated intervals. For longer intervals, the authors state "there was no need for the animals to maintain the variation of saccade timing within several hundreds of milliseconds". This begs the question of why there is there a "need" at all to maintain variation in timing. Perhaps inherent variation allows us to explore different intervals in a world with constantly changing temporal contingencies – but if so, why would we relax that exploratory bent for supra-second intervals?

We agree with the reviewer in that there was no need for the animals to maintain variation of saccade timing in our behavioral paradigm. We have removed the relevant sentences from the text in the Discussion section.

Stepping back, there were two observations behind the claim that the cerebellum is responsible for stochastic variation. First, for the medium and long intervals, if trials are grouped according to precise movement time, the neuronal responses in the Cb appear to diverge a few hundred ms earlier than those of the striatum (Figure 5). This earlier divergence is based (per force) on a statistical criterion – when significant differences first arise based on the results of repeated-measures ANOVA. This is not a fair comparison between caudate and Cb. Average firing rates in caudate are far lower than in Cb dentate firing (Figure 3C-D). Moreover, I'm not sure about dentate nucleus firing statistics, but firing patterns in the Cb can be highly regular (e.g., Purkinje cells), which would reduce variance in spike counts for a given mean firing rate. For both these reasons, it might just be easier to DETECT the ANOVA-based divergence point for the cerebellar neurons than the caudate neurons, whereas we don't know how the brain actually pools signals from these two areas. I'm not exactly sure how to address this problem, but the authors could try to repeat their analysis with a subset of data matched for mean firing rates from the two areas (if possible). It would also be helpful to at least examine the coefficient of variation of firing rates between the two areas.

In response to this comment, we now report the coefficient of variation of baseline firing rates for the two areas (subsection “Neuronal correlates of trial-by-trial variation in self-timing”). As the reviewer pointed out, the baseline firing was more regular and the magnitude of preparatory activity was much greater in the cerebellum than the caudate nucleus. It is worth noting that the analyses in Figure 5 were based on the normalized activity for individual neurons, and that the repeated measures ANOVAs were performed on the spike density profiles derived from three latency groups. This procedure was optimal to detect the earliest diverging point of mean firing rate across groups, while the variation of baseline activity in each trial did not greatly confound the present results. We have added some lines to clarify this point (subsection “Neuronal correlates of trial-by-trial variation in self-timing”). In addition, we now also report that the SDs of normalized baseline activity for the spike density profiles used for the analysis were not different between the recording sites in Figure 5 legend.

Second, the authors conclude the description of the inactivation experiment (end of Results) with the statement that stochastic variation "might be controlled by signals in the cerebellum, but not in the striatum" – but this does not seem warranted from the inactivation experiment. The clearest result is that inactivation of Cb affected mean movement time most for short intervals while striatum affected mean movement time most for medium and long times. With respect to *variation* in latency, the effects of Cb inactivation were inconsistent at best. In addition, the largest change in movement-time variation with Cb inactivation was for short intervals, but this also caused the largest shift in mean movement latency of any area/interval/experiment, which might be expected to automatically increase the variation in latency (e.g., as per scalar expectancy theory of interval timing). But more generally, what is the hypothesized effect of inactivation on latency variation? Do the authors posit that the cerebellum controls precision, and thus that Cb inactivation would erode that precision? Is that interpretation not at odds with the results in Figure 5? Or do the authors hypothesize that the Cb adds noise to the timing process, in which case Cb inactivation might reduce variation?

We agree with the reviewer. In the revised manuscript, we have removed the claim that the cerebellum may control the stochastic variation. Instead, we now mention that the stochastic variation started earlier in the Cb than the striatum, and that the Cb might play a role in the fine adjustment of timing.

In the end, the clearest result in the paper is the differential roles for the caudate and Cb for long and short intervals, respectively, supported by both electrophysiological recording and inactivation experiments. This alone is a nice contribution. The stochastic variation claims are not strongly supported (either empirically or theoretically). Of course, the authors could present the "divergence" argument in Figure 5, but in a more conservative manner, pointing out the difficulty in statistically distinguishing divergence points between neuronal populations with different firing rates and potentially different firing statistics. But I don't think that the cerebellar stochastic variation claims should be featured prominently in the paper (for example, the abstract should not conclude with that claim).

We appreciate these positive comments. In the revised manuscript, we only report the fact that the stochastic variation started earlier in the cerebellum but have eliminated the argument about its role. Accordingly, we have also modified the Abstract.

We also ask that you double check the new supplement 1 to Figure 3 (saccade-aligned responses). The SD of cue times seems narrow given the wide distributions of movement times in Figure 1B. Please verify that the figure is correct.

We have verified that they are correct. In both Figure 3 and Figure 3—figure supplement 1, the error bar indicates the inter-experimental variability and plots the SD of means for individual recording sessions. We now clarify this point in figure legends.